# The Problem of Vitamin D Scarcity: Cultural and Genetic Solutions by Indigenous Arctic and Tropical Peoples

**DOI:** 10.3390/nu14194071

**Published:** 2022-09-30

**Authors:** Peter Frost

**Affiliations:** Anthropology, Université Laval, Quebec City, QC G1V 0A6, Canada; peter_frost61z@telus.net

**Keywords:** Arctic, culture, genetics, Inuit, Sámi, Samoyed, Tropics, UVB, vitamin D

## Abstract

Vitamin D metabolism differs among human populations because our species has adapted to different natural and cultural environments. Two environments are particularly difficult for the production of vitamin D by the skin: the Arctic, where the skin receives little solar UVB over the year; and the Tropics, where the skin is highly melanized and blocks UVB. In both cases, natural selection has favored the survival of those individuals who use vitamin D more efficiently or have some kind of workaround that ensures sufficient uptake of calcium and other essential minerals from food passing through the intestines. Vitamin D scarcity has either cultural or genetic solutions. Cultural solutions include consumption of meat in a raw or boiled state and extended breastfeeding of children. Genetic solutions include higher uptake of calcium from the intestines, higher rate of conversion of vitamin D to its most active form, stronger binding of vitamin D to carrier proteins in the bloodstream, and greater use of alternative metabolic pathways for calcium uptake. Because their bodies use vitamin D more sparingly, indigenous Arctic and Tropical peoples can be misdiagnosed with vitamin D deficiency and wrongly prescribed dietary supplements that may push their vitamin D level over the threshold of toxicity.

## 1. Introduction

Different human populations differ in the ways they produce and use vitamin D. This is because our species has adapted to different natural and cultural environments. Two environments are particularly difficult for the production of vitamin D by the skin:The Arctic, where sunlight is weak in summer and absent in winter. The skin thus receives little of the solar UVB that helps produce vitamin D. This is the environment of people indigenous to the North American and Eurasian Arctic, notably the Aleut, the Inuit (Eskimos), the Sámi (Lapps), the Samoyeds, the Khanty-Mansi, the Yakuts, the Yukaghir, the Tungus, the Koryaks, and the Chukchi.The Tropics, where intense sunlight has favored the survival of individuals with highly melanized skin that blocks UVB. This is notably the environment of people indigenous to sub-Saharan Africa, southern India, Australia, Papua New Guinea, and Melanesia.

Without enough vitamin D, you cannot absorb enough calcium, phosphorus, magnesium, and zinc from food passing through your intestines. Your bones will be insufficiently mineralized and become soft and weak, the eventual outcome being rickets if you are a child and osteomalacia or osteoporosis if you are an adult. Other adverse outcomes may include impairment of the immune system and increased risk of diabetes and certain cancers.

When vitamin D is scarce, natural selection favors the survival of those individuals who use it more efficiently or have some kind of workaround that ensures sufficient uptake of calcium and other essential minerals. This selection can occur because individuals already vary in different aspects of vitamin D metabolism. First of all, they vary in the normal serum level of vitamin D. That level has an estimated heritability of 20 to 85%, the wide range of estimates being due to difficulties in controlling for age, sex, and season. Heritability is higher for levels of the most active form of vitamin D, 1,25(OH)2D, than for those of the most common form, 25(OH)D [1,2]. Individuals may vary in other aspects of vitamin D metabolism: production in the skin; degradation in the skin; transportation via the bloodstream; and uptake of calcium and other essential minerals from the intestines. There is thus a range of possible genetic solutions to the problem of vitamin D scarcity.

There are also cultural solutions; that is, learned ways of using vitamin D more efficiently or more sparingly. When a population enters a new environment, the problems of adaptation are solved by cultural practices; you learn to do the best you can with what you have. Over time, natural selection will then create genetic solutions; in this case, changes to the metabolism of vitamin D (Figure 1). Genetic evolution tends to lag behind cultural evolution, but not always. A simple genetic change may spread rapidly through a population in a matter of generations, whereas a cultural practice may take a long time to overcome resistance to new ways of doing things [3,4].

## 2. Vitamin D Scarcity in Arctic Peoples

At high latitudes, where sunlight is weak, the skin cannot produce sufficient vitamin D. It is untrue, as some claim, that Arctic peoples were able to make up the difference from dietary sources of vitamin D, notably fatty fish and certain marine mammals. According to a study of Inuit in northern Canada, the median daily intake of vitamin D was 5.13 ± 5.34 µg for Inuit on a traditional diet and 3.5 ± 3.22 µg for those on a non-traditional diet. The reference intake of 10 µg per day was not reached even by members of the first group, who daily consumed at least 300 g of game meat (Arctic char, trout, caribou, seal, musk ox, goose, ptarmigan) [5].

### 2.1. Cultural Solutions

Although Arctic peoples cannot obtain sufficient vitamin D from their skin or their diet, it is only in recent times that they have begun to suffer from rickets and other adverse effects of vitamin D deficiency. Those adverse effects were prevented by certain aspects of their traditional diet: consumption of meat in a raw or boiled state; extended breastfeeding of children for at least two years after birth; and no consumption of cereals.

#### 2.1.1. Consumption of Raw or Boiled Meat

Meat consumption reduces the risk of rickets and osteomalacia independently of the meat’s vitamin D content, with no further risk reduction above 60 g of meat per day [6]. The anti-rickets component of meat remains unknown, although meat protein has been ruled out [7]. As shown below, it seems to be a nutritional cofactor that survives boiling, but not cooking.

Arctic peoples traditionally ate large quantities of meat. The Inuit, for instance, subsisted largely on raw or boiled meat from fish, marine mammals, and land mammals [8,9,10,11]. Although Arctic peoples had to eat meat, there being few other food sources, they were not forced to eat it in a raw or boiled state. Yet such consumption was the norm: “The preparation of Eskimo food is simple. Most meat is eaten partly boiled and the remainder raw” [11] (p. 462). That norm was unknown to, and scorned by, indigenous peoples who lived farther south, such as the Amerindians of temperate North America. “If the Indians are to be believed, their hatred [of the Inuit] springs from another source. They do not know how to forgive the Esquimaux the crime of eating raw fish” [12] (p. 9). The name “Eskimo” is alleged to have come from an Algonkian term meaning “eaters of raw meat” and has been replaced by “Inuit” because of its negative connotation [10] (p. 69), [13] (p. 101).

A 19th century observer noted the same dietary practice among the Samoyed peoples of northern Russia. Viewing it as a circumpolar adaptation, he concluded: “There must be something in this universal craving in the Arctic regions for the freshest of meat and for vitalising blood, and I attribute the immunity of both Eskimo and Samoyad from scurvy to their persistent use of this coarse but vitalising food” [14] (p. 404). Like the Inuit, the Samoyed were once called “eaters of raw meat” [14] (p. 391). 

Although cooking makes meat easier to digest and tastier, it appears to destroy certain nutritional cofactors that may be difficult to obtain from other food sources, particularly in the Arctic [15]. Arctic peoples may not understand the why or the how, but they do understand the health benefits of uncooked meat: “Inuit emphasize how eating raw seal meat or hunting at the floe edge produces healthy bodies and intelligent minds … One Inuit man, who had spent much of his youth at a hunting camp and later moved to live in Iqaluit, said that his town diet of white foods made him weak, lazy, and ill equipped to deal with the strength and stamina needed to live off the land” [16] (p. 247).

The above observations are consistent with earlier examples that show scurvy was confined to those Inuit who, under Western influence, were now cooking their meat [15]. According to Arctic explorer Elisha Kent Kane (1820–1857), the natives of South Greenland believed that raw meat gave them the energy to go on long journeys, for “fire would ruin the curt, pithy expression of vitality which belongs to its uncooked juices” [17] (pp. 274–275). This unknown but vital cofactor seems to be vulnerable to oxidation under conditions of high temperature and low water content.

#### 2.1.2. Extended Breastfeeding

A mother can reduce her child’s need for vitamin D by breastfeeding for a longer period after birth. Breast milk is rich in a phosphoprotein called β-casein, which helps keep calcium soluble during digestion and contributes to its bioavailability [18]. A similar role is played in milk by phosphate, citrate, and other caseins [19]. Consequently, breastfeeding reduces a child’s need for vitamin D.

Inuit mothers would breastfeed their children for at least two years after birth, a custom noted by early 20th century observers and linked to the lack of rickets:

“Babies are wholly breast-fed until the end of the 2nd year when meat is added to their diet. Children often nurse irregularly until the age of 4 or 6”.[11] (p. 463)

“Among these primitive, carnivorous people there is neither scurvy nor rickets. Children are nursed for four and not infrequently six years”.[20] (p. 1560)

“The reason, however, for the failure to find rickets at the above-mentioned places appears to be that infants are nursed for as much as two years and more, and the fact that the vitamin D content of seal oil is equal to that of the best cod-liver oil. According to Dr. Urquhart, infants are nursed for very long periods of time amongst the Western Arctic Eskimos also”.[21] (p. 494)

The same explanation was offered for the lack of rickets among the Sámi of northern Scandinavia. An early 20th century observer found only one case of rickets among Sámi children despite the short summer. Breastfeeding, however, lasted from 10.5 months to 4 years. Finnish and Swedish children in the same region were breastfed for only 3 to 4 months and had a high incidence of rickets, despite their living under similar climatic conditions [22].

Extended breastfeeding for two to three years is also reported from the Khanty of western Siberia, although the stated reason is to postpone the next pregnancy [23] (p. 152).

At least among the Inuit, extended breastfeeding did not become the norm until late in prehistory. An analysis of 2000-year-old Inuit remains from Chukotka shows that only half of the individuals were breastfed beyond the age of two years (breastmilk leaves a discernible isotope signature in developing bones). The proportion was 40% at a site farther south, near Lake Baikal, dated to 7500–6700 BP [24] (pp. 57–58). Extended breastfeeding seems to have become established through a slow process of cultural evolution: late-weaning mothers gradually increased in number because their daughters were more likely to survive to adulthood and become late-weaning mothers themselves.

#### 2.1.3. No Consumption of Cereals

Finally, the need for vitamin D can be reduced by not eating cereals. Consumption of cereals increases the risk of rickets independently of the body’s supply of vitamin D [7,25,26,27,28]. The actual risk factor seems to be phytic acid, which is found in the hulls of nuts, seeds, and grains and is most often ingested from store-bought bread. By binding to minerals to form insoluble precipitates, it reduces the body’s supply of usable essential minerals, like calcium, and increases the body’s need for vitamin D. That effect has been shown by animal and human studies. When puppies are fed oatmeal, they develop rickets in proportion to its phytic acid content [25]. When humans are fed a 92% flour diet, they absorb less calcium, magnesium, phosphorus, and potassium than they do on a 69% flour diet. When the diet is 92% flour, the loss of usable calcium cannot be counteracted by adding vitamin D [26]. Calcium, magnesium, and phosphorus are absorbed in inverse proportion to the phytic acid content of the flour diet [27].

Cereals were unknown in the Arctic before European contact. They were first consumed in northern Canada by the Nunatsiavummiut of Labrador, and it was among them that rickets first appeared. By the 1920s the disease had reached a high incidence, which a visiting physician blamed on certain dietary changes, particularly a shift from raw meat to cereal products: “Wood is abundant, so they cook their meat. The Moravian missions and the Hudson Bay Company, with the best of intentions, take their furs and sell them provisions—dried potatoes, flour, canned goods, cereals and cereal products—and their fare consists largely of these staples. Consequently, scurvy, rickets and combinations of the two are universal” [20] (p. 1560).

Although cereals were absent from the Arctic diet until recent times, their absence cannot be considered a true cultural solution to the problem of vitamin D scarcity. It was simply an unavoidable circumstance of Arctic life.

### 2.2. Genetic Solutions

Arctic peoples have coped with vitamin D scarcity through changes not only to their diet but also to their physiology. The latter changes have taken several forms: higher uptake of calcium from the intestines; higher rate of conversion of vitamin D from its common form to its most active form; and stronger binding of vitamin D to carrier proteins in the bloodstream, as well as higher serum levels of those carrier proteins.

#### 2.2.1. Higher Calcium Uptake

A study from northern Canada shows that Inuit children have less need for calcium in their diet and thus less need for vitamin D. Whereas the recommended daily calcium intake is 800 mg for children 4 to 8 years old and 1300 mg for those 9 years old or more, Inuit children on a traditional diet ingest only 20 mg per day. They also excrete excess calcium at an unusually high rate despite their low calcium intake. The explanation seems to be that their vitamin D receptor (bb genotype) is associated with a higher uptake of calcium from food passing through the intestines. The authors concluded: “Dietary calcium intakes based on North American guidelines may therefore result in iatrogenic hypercalciuria and renal damage” [29]. In a study at Inuvik, in northern Canada, both Inuit and Amerindian newborns had 25(OH)D levels that were only 67% of their mothers’, while showing no signs of vitamin D deficiency and having calcium levels significantly higher than their mothers’ [30].

A higher uptake of calcium may be typical of Amerindian peoples in general. Their ancestors in Beringia had to adapt to high latitudes that were less conducive to production of vitamin D, and some, like those of Inuvik, still live at high latitudes. Such an adaptation may explain why Amerindian women have a higher bone mineral density and a lower risk of hip fractures before menopause than do women of European descent [31,32]. It is also possible, as will be discussed further, that their higher bone mineral density is due to a greater capacity to absorb calcium from food independently of vitamin D.

#### 2.2.2. Higher Rate of Conversion

A study of adult Greenland Inuit has shown that they evolved at least two genetic solutions to the problem of vitamin D scarcity. In addition to having a different calcium metabolism, as indicated by a lower set-point for the calcium-regulated release of parathyroid hormone, they also have a higher rate of conversion of vitamin D from its common form to its most active form. Serum levels of 1,25(OH)2D are thus higher in Greenland Inuit than in Danes. “Due to a low endogenous 25OHD production in Greenlanders … evolution may have selected individuals with a relatively high 25-hydroxyvitamin D1α-hydroxylase activity” [33] (p. 261).

#### 2.2.3. Stronger Binding by Carrier Proteins

Among the indigenous peoples of Arctic and sub-Arctic Eurasia, vitamin D tends to be bound more strongly by the proteins that transport it via the bloodstream. There are also higher serum levels of the same carrier proteins. The main one is produced by the gene *GC*, which exists in two variants: the T variant, which is much less frequent in the northeast and center of Siberia than in the south and west; and the G variant, which is conversely much more frequent in the northeast and center than in the south and west. The G variant is associated with increased transportation of vitamin D via the bloodstream; it has thus been favored by natural selection in populations that are farther north and less able to produce vitamin D. This difference in geographic distribution cannot be explained by different levels of European admixture, since East Asians have very little European admixture and yet resemble southern and western Siberians in having the same high frequency of the T variant [34]. Natural selection seems to have likewise favored the G variant in Arctic and sub-Arctic North America, the population frequency being 71% among the Dene peoples of northern Canada. Moreover, 90% of them have the Gc1 genotype, which is associated with higher serum levels of carrier proteins and stronger binding to vitamin D [35].

There may be other genetic variants that strengthen carrier protein binding, as well as others that increase the efficiency of vitamin D metabolism in other ways. When the genomes of several populations in northern Russia were examined for signals of natural selection, the strongest signal was found at two genes: *SLC37A2* and *PKNOX2*. The first gene is expressed when vitamin D_3_ is present in peripheral blood cells, and the second may have a role in vitamin D metabolism because it lies at the same locus on the opposite strand of DNA [36].

## 3. Vitamin D Scarcity in Tropical Peoples

It may seem strange that less vitamin D is produced in the skin of people from the Tropics, where sunlight is so intense. The same intense sunlight, however, has also selected for highly melanized skin that blocks UVB and thus limits production of vitamin D. That blocking effect has been shown in Americans with varying degrees of African ancestry. With each 10% increase in African ancestry, the serum 25(OH)D level decreases by 2.5 to 2.75 nmol/L [37]. Levels are lower than 50 nmol/L during the winter in 53 to 76% of African Americans who live in the southern states [38].

A similar situation prevails in other dark-skinned peoples, even those who have remained in the Tropics and are regularly exposed to intense solar UV. In south India, at 13.4° N, a study found that 44% of the men and 70% of the women had 25(OH)D levels lower than 50 nmol/L. They were “agricultural workers starting their day at 0800 and working outdoors until 1700 with their face, chest, back, legs, arms, and forearms exposed to sunlight” [39]. That finding is corroborated by another Indian study, which found levels higher than or equal to 50 nmol/L in only 31.5% of the participants, who had nonetheless been exposed to the sun for 5 h every day [40]. Two studies from the Middle East have reported levels lower than 50 nmol/L in 91% of healthy athletes [41] and lower than 25 nmol/L in 35%, 45%, 53%, and 50%, of Saudi, Jordanian, Egyptian, and other male university students in Riyadh [42]. Finally, a meta-analysis concluded that 25(OH)D levels are significantly higher in people of European origin than in those of non-European origin. The latter have low levels regardless of latitude [43].

### 3.1. Cultural Solutions

Dark-skinned peoples are thus faced with a problem of vitamin D scarcity. Unlike Arctic peoples, they do not solve it by means of cultural practices, at least not at the present time. Meat provides a small proportion of the caloric intake in sub-Saharan Africa and is normally cooked, and cereals like sorghum and millet have long been consumed despite their high levels of phytic acid [44,45]. On the other hand, extended breastfeeding may have been formerly common [46] (p. 15), [47] (pp. 6–7).

Cultural solutions are unlikely for both theoretical and evidentiary reasons. First, humans and their hominin forbears have lived much longer in the Tropics than in the Arctic—millions of years longer. Natural selection has thus had considerable time to replace the initial cultural solutions with genetic ones. Second, cultural solutions, if still needed, would surely include a prohibition against cooking of meat, a widespread dietary rule among Arctic peoples.

### 3.2. Genetic Solutions

A genetic adaptation is often a hardwiring of something initially imposed by circumstance. In this case, dark-skinned peoples originally had low levels of vitamin D because their skin is less easily penetrated by UVB. Today, there is a second constraint, a homeostatic mechanism that resists any effort to increase the body’s supply of vitamin D. This has been shown in Americans with varying degrees of African ancestry. Both sunlight and diet are 46% less effective in raising the 25(OH)D level of individuals with high African ancestry [37].

The tendency toward homeostasis suggests that a low 25(OH)D level is sufficient for African Americans. Indeed, few of them show signs of vitamin D deficiency. They have “a lower prevalence of osteoporosis, a lower incidence of fractures and a higher bone mineral density than white Americans, who generally exhibit a much more favorable vitamin D status” [48]. Among women 65 years of age, the risk of a hip fracture by age 80 is only 4% for African Americans versus 11% for European Americans [32,38]. Among teenage girls, calcium retention, bone formation rates, and calcium absorption efficiency, are higher in African Americans than in European Americans [49]. A similar picture emerges from a survey of East African immigrant children in Australia, 87% of whom had 25(OH)D levels lower than 50 nmol/L and 44% lower than 25 nmol/L. None had rickets, the usual result of vitamin D deficiency in children [50]. A review of the literature concludes that “the intake [of calcium] needed to ensure optimal skeletal status is lower in Blacks than in the other racial groups [Europeans and East Asians]” [51] (p. 153).

A difference in metabolism is further shown by a finding that bone mineral density correlates with serum 25(OH)D in European Americans but not in African Americans or Hispanic Americans [52,53,54]. The non-correlation in Hispanic Americans may seem puzzling, although the research participants could have been Puerto Ricans and Dominicans with substantial African ancestry. It is also possible that vitamin D and calcium are metabolized differently even in sub-tropical peoples with medium skin color. This is suggested by a study of Middle Eastern athletes, who showed no correlation between bone mineral density and serum 25(OH)D. Nor did any of them have stress fractures, despite low 25(OH)D levels [41].

How exactly do dark-skinned peoples use vitamin D more efficiently and more sparingly? There seem to be two ways: a higher rate of conversion of vitamin D to its most active form; and, perhaps, a greater use of alternative metabolic pathways for calcium uptake.

#### 3.2.1. Higher Rate of Conversion

Although a single UVB exposure causes the skin to produce less vitamin D_3_ in African Americans than in European Americans, the difference decreases after liver hydroxylation to 25(OH)D and is gone after kidney hydroxylation to 1,25(OH)2D. The most active form of vitamin D is thus produced at a constant rate regardless of how dark the skin is [55,56].

#### 3.2.2. Greater Use of Alternative Metabolic Pathways

Some calcium is absorbed into the body through pathways that function independently of vitamin D. Those pathways include passive diffusion of calcium ions [57,58] and transportation of calcium in the form of calcium chelate [59]. There is also suggestive evidence that bile salts, lactose, and prolactin assist calcium uptake independently of vitamin D [58]. Natural selection may have expanded such alternative pathways in populations that lack vitamin D.

## 4. Discussion

Further research is needed on the vitamin D metabolism of Arctic and dark-skinned peoples. Current standards are based on the vitamin D intake and serum levels of Europeans and North Americans, the assumption being that what holds true for them should also hold true for humanity in general. Unfortunately, that is not always so. Had the data come initially from dark-skinned or Arctic participants, the standards would be quite different today, and we would be diagnosing Europeans and North Americans with hypervitaminosis.

Current standards are all the more problematic because 25(OH)D levels are beneficial within a relatively narrow range, essentially, 40 nmol/L to 100 nmol/L in light-skinned humans from the temperate zone. Levels outside that range are associated with higher mortality:The total mortality rate is about 50% greater among men whose 25(OH)D levels are lower than 46 nmol/L or higher than 98 nmol/L [60];The risk of prostate cancer is significantly greater at levels lower than 40 nmol/L or higher than 60 nmol/L [61,62];For endometrial, esophageal, gastric, kidney, non-Hodgkin’s lymphoma, pancreatic, and ovarian cancer, the mortality rate is significantly greater at levels lower than 45 nmol/L or higher than 124 nmol/L [63];The risk of pancreatic cancer is significantly greater at levels higher than 100 nmol/L [64];The risk of cardiovascular disease is significantly greater at levels lower than 50 nmol/L or higher than 62.5 nmol/L, and the mortality rate for all causes is significantly greater at levels higher than 122.5 nmol/L [65].

Impairment of cognition may likewise follow the same U-shaped response curve. In mice, poorer maze performance is associated as much with high 25(OH)D levels as with low ones [66]. Perhaps most worrisome, other mouse studies have shown a similar response curve for the aging process; premature aging is associated with levels that are either too low or too high [61,62].

When vitamin D is used more sparingly in the body, it should have beneficial effects at lower serum levels. This is the case with the risk of tuberculosis, which remains low among Africans at progressively lower levels of 25(OH)D, whereas Asians have a progressively higher risk across the same range of levels [67]. If the beneficial range of levels is lower, the upper bound of that range should also be lower. This is the case with vitamin D supplementation, which fails to reduce bone loss or increase bone turnover in postmenopausal African American women at a pre-treatment level of only 47 nmol/L [54].

Finally, if the upper bound of the beneficial range is lower, the threshold of toxicity should also be lower. This is the case with formation of calcified atherosclerotic plaque, which becomes progressively worse among African Americans across a range of 25(OH)D levels that is considered beneficial for European Americans [53]. Individuals with dark skin, or from the Arctic, may thus be misdiagnosed as vitamin D deficient, wrongly prescribed dietary supplements, and thereby pushed over the threshold of toxicity [68,69,70,71,72]. Keep in mind that vitamin D is fat-soluble and thus accumulates in the body. Unlike vitamin C, it cannot be removed from the body through urination if too much is ingested.

Two studies point to the possible risks of dietary supplementation for Arctic and dark-skinned peoples. One study showed an apparent increase in the risk of tuberculosis when Amerindians (Dene) in northern Canada were given vitamin D supplementation. They experienced a significant decrease in their levels of LL-37, an antimicrobial peptide that helps the body defend itself against tuberculosis. Counterintuitively, the decrease was confined to those Dene whose carrier proteins bind more weakly to vitamin D [73]. The other study showed an increase between 2011 and 2016 in vitamin D levels and hypervitaminosis D among patients at an Indian medical institute, apparently due to over-supplementation. Similar increases in vitamin D levels have been reported from Ireland, England, Canada, and Australia, but, unlike India, those countries have shown no corresponding increase in reports of hypervitaminosis D [74].

Only one research team has looked into the possibility that the beneficial range of vitamin D levels is different in different populations. According to a study of Inuit and Danes in Greenland, the risk of active tuberculosis is greater at levels lower than 75 nmol/L or higher than 140 nmol/L. The authors concluded that “the OR did not change considerably when adjusting for ethnicity.” However, the Danish participants were few in number, being only eight of the 72 patients and 19 of the 72 controls [75]. The Greenland Inuit are also highly admixed, with 58% of their paternal ancestry being European [76].

Until more is known about how Arctic and dark-skinned peoples metabolize vitamin D, they should not be targeted for dietary supplementation, particularly in the case of individuals who are healthy and show no signs of vitamin D deficiency other than low serum levels. Arctic communities should also be encouraged to return to a more traditional diet with less consumption of cereals and more consumption of local meat, as is already being attempted through the opening of “country food markets” in Greenland and northern Canada [77].

## Figures and Tables

**Figure 1 nutrients-14-04071-f001:**
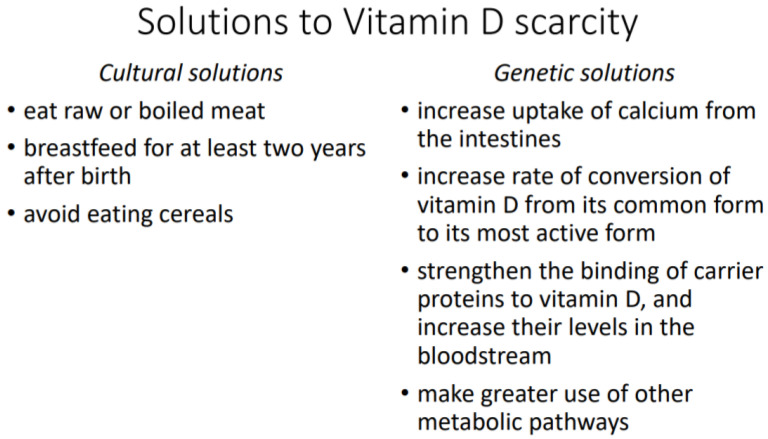
Solutions to vitamin D scarcity.

## Data Availability

Not applicable.

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
