# Peer review of "The Problem of Vitamin D Scarcity: Cultural and Genetic Solutions by Indigenous Arctic and Tropical Peoples"

_nutrients, 2022, doi:10.3390/nu14194071_

Round 1

Reviewer 1 Report

Thank you for your contribution to our journal. 

This is an interesting paper to understand the vitamin D deficiency and

adaptation by cultural, diet, lifestyle in Arctic and Tropics.

Dr. Holic, in Boston University, insisted and I want to have one question for the increased risk of Tbc after D supplementation, even though proper serum D level has been known the protection role of Tbc, etc, cathelicidin.  

Author Response

Yes, I have tried to describe how cultural adaptations tend to pave the way for genetic adaptations in Arctic and Tropical populations.

With respect to the risk of tuberculosis, the beneficial range of vitamin D levels seems to be lower in Africans than in Asians (see p. 8 of my paper). In a study of Inuit and Danish Greenlanders, the risk of active tuberculosis is greater at levels lower than 75 nmol/L or higher than 140 nmol/L (see p. 8). It is unclear whether this beneficial range differs between Danes and Inuit. 

Reviewer 2 Report

The review proposed by Prof. Frost offers an unprecedented and stimulating point of view in the field of Vitamin D metabolism. However, in this reviewer’s opinion, the anthropological aspects covered in this review seem to overcome the clinical and nutritional aspects of the Vitamin D use in many paragraphs of the manuscript, likely linked to basic anthropological background of the Author.

Therefore, I recommend some minor revisions listed below, but I prefer to re-submit to the editorial board the final decision to accept for publication in Nutrients a text with a prevalent anthropological topic.

1)      I recommend choosing a title more suited to the content of the review, highlighting the cultural or genetic causes of plasma vitamin D differences in the two populations.

2)      Which can be the cofactor contained in raw meat that could act independently of the meat's vitamin D content and what could be the mechanism of action? Please specify or try to provide a hypothesis based on available scientific evidence.

3)      Please clarify in paragraph "2.2.1. Higher calcium uptake" which are the possible mechanisms justifying the excessive excretion of renal calcium despite the low serum calcium levels.

4)      It is advisable to expand and detail paragraph 2.2.2.

5)      As reported for Arctic people, it is advisable to search and indicate more cultural aspects for vitamin D scarcity in Tropical people in the “Cultural solutions” paragraph.

6)      In the text the Author mainly refers to rickets, but it would be interesting to analyse other pathologies associated with the metabolism of calcium and vitamin D, such as osteoporosis, and how their incidence and prevalence varies in the two populations.

7)      Considering the cultural and genetic differences in vitamin D metabolism in the two populations, have different cut-offs for plasma vitamin D levels been proposed to indicate deficiency or sufficient conditions? Please specify.

8)      A figure that explains the main cultural and genetic differences between populations could be useful to facilitate the understanding of the text.

Author Response

The review proposed by Prof. Frost offers an unprecedented and stimulating point of view in the field of Vitamin D metabolism. However, in this reviewer’s opinion, the anthropological aspects covered in this review seem to overcome the clinical and nutritional aspects of the Vitamin D use in many paragraphs of the manuscript, likely linked to basic anthropological background of the Author.

Therefore, I recommend some minor revisions listed below, but I prefer to re-submit to the editorial board the final decision to accept for publication in Nutrients a text with a prevalent anthropological topic.

R: Thank you for your detailed comments. 

1)      I recommend choosing a title more suited to the content of the review, highlighting the cultural or genetic causes of plasma vitamin D differences in the two populations.

R: The title has been changed to: The Problem of Vitamin D Scarcity: Cultural and Genetic Solutions by Indigenous Arctic and Tropical Peoples.

2)      Which can be the cofactor contained in raw meat that could act independently of the meat's vitamin D content and what could be the mechanism of action? Please specify or try to provide a hypothesis based on available scientific evidence.

R: The nutritional cofactor is a molecule that survives boiling but not cooking. Specifically, it seems to be vulnerable to oxidation under conditions of high temperature and low water content. This has been added to the text.

3)      Please clarify in paragraph "2.2.1. Higher calcium uptake" which are the possible mechanisms justifying the excessive excretion of renal calcium despite the low serum calcium levels.

R: Sorry, that was a mistake. I have replaced "higher calcium uptake" with "low calcium intake." The Inuit ingest low levels of calcium from their diet, but they seem to be more effective in absorbing calcium from food passing through the intestine.

4)      It is advisable to expand and detail paragraph 2.2.2.

R: Paragraph 2.2.2 has been expanded.

5)      As reported for Arctic people, it is advisable to search and indicate more cultural aspects for vitamin D scarcity in Tropical people in the “Cultural solutions” paragraph.

R: In my opinion, there are few if any cultural adaptations to vitamin D scarcity in tropical peoples, given the long length of time that humans have lived in the Tropics. I have explained this point in the text: 

"Cultural solutions are unlikely for both theoretical and evidentiary reasons. First, humans and their hominin forbears have lived much longer in the Tropics than in the Arctic—millions of years longer. Natural selection has thus had considerable time to replace the initial cultural solutions with genetic ones. Second, cultural solutions, if still needed, would surely include a prohibition against cooking of meat, a widespread dietary rule among Arctic peoples."

6)      In the text the Author mainly refers to rickets, but it would be interesting to analyse other pathologies associated with the metabolism of calcium and vitamin D, such as osteoporosis, and how their incidence and prevalence varies in the two populations.

R: There is not much in the literature about population differences in osteoporosis. I now discuss two studies on this point (hip fractures) with respect to Amerindians on page 5 and three studies on this point with respect to African Americans on page 6.

7)      Considering the cultural and genetic differences in vitamin D metabolism in the two populations, have different cut-offs for plasma vitamin D levels been proposed to indicate deficiency or sufficient conditions? Please specify.

R: Unfortunately, no population-specific guidelines on vitamin D have been proposed by any international or national organization. This may change.

8)      A figure that explains the main cultural and genetic differences between populations could be useful to facilitate the understanding of the text.

R: I have prepared a figure, for insertion on page 2.

Round 2

Reviewer 2 Report

The authors have adequately fulfilled this reviewer's suggestions which were proposed in the first revision of the manuscript